# Association between Antidepressant Treatment during Pregnancy and Postpartum Self-Harm Ideation in Women with Psychiatric Disorders: A Cross-Sectional, Multinational Study

**DOI:** 10.3390/ijerph18010046

**Published:** 2020-12-23

**Authors:** Jennifer Vallee, Yih Wong, Eline Mannino, Hedvig Nordeng, Angela Lupattelli

**Affiliations:** 1PharmacoEpidemiology and Drug Safety Research Group, Department of Pharmacy, Faculty of Mathematics and Natural Sciences, University of Oslo, 0316 Oslo, Norway; jennifer.vallee@admin.uio.no (J.V.); yih.wong@studmed.uio.no (Y.W.); elinemannino@gmail.com (E.M.); h.m.e.nordeng@farmasi.uio.no (H.N.); 2Department of Child Health and Development, Norwegian Institute of Public Health, 0213 Oslo, Norway

**Keywords:** antidepressants, pharmacotherapy, pregnancy, postpartum, self-harm ideation, web-based

## Abstract

This study sought to estimate whether there is a preventative association between antidepressants during pregnancy and postpartum self-harm ideation (SHI), as this knowledge is to date unknown. Using the Multinational Medication Use in Pregnancy Study, we included a sample of mothers who were in the five weeks to one year postpartum period at the time of questionnaire completion, and reported preexisting or new onset depression and/or anxiety during pregnancy (*n* = 187). Frequency of postpartum SHI (‘often/sometimes’ = frequent, ‘hardly ever’ = sporadic, ‘never’) was measured via the Edinburgh Postnatal Depression Scale (EPDS) item 10, which reads “*The thought of harming myself has occurred to me*”. Mothers reported their antidepressant use in pregnancy retrospectively. Overall, 52.9% of women took an antidepressant during pregnancy. Frequent SHI postpartum was reported by 15.2% of non-medicated women and 22.0% of women on past antidepressant treatment in pregnancy; this proportion was higher following a single trimester treatment compared to three trimesters (36.3% versus 18.0%). There was no preventative association of antidepressant treatment in pregnancy on reporting frequent SHI postpartum (weighted RR: 1.90, 95% CI: 0.79, 4.56), relative to never/hardly ever SHI. In a population of women with antenatal depression/anxiety, there was no preventative association between past antidepressant treatment in pregnancy and reporting frequent SHI in the postpartum year. This analysis is only a first step in providing evidence to inform psychiatric disorder treatment decisions for pregnant women.

## 1. Introduction

During pregnancy, between 2% and 6% of women take antidepressants to treat perinatal psychiatric disorders, mainly depression and anxiety, and their use appears to be trending upward since the 1990s [1,2,3]. Women with an antenatal mental illness are at high risk for sustained illness during the perinatal period and/or for a relapse up to one year postpartum [4,5]. Despite this, it is unclear to what extent antidepressant treatment in pregnancy could reduce the risk of severe mental health outcomes after childbirth, including self-harm and attempted suicide [6,7].

Self-harm ideation (SHI) involves thoughts of poisoning or inflicting injuries on oneself, with or without fatal intent, and is an important risk factor for repeated SHI episodes and suicide [8,9]. Although the prevalence of SHI differs depending on the mental illness considered, case ascertainment methods, and geography, up to one in five women with a psychiatric disorder may experience an event of SHI in the postpartum period [10]. This high estimate is alarming as SHI causes direct harm to the mother, may impair her ability to care for the child, and disrupts the welfare of the family as a whole [11]. In the UK in 2015–2017 the rate of maternal perinatal death from suicide was 0.6 for 100,000 maternities [12]. Consistent findings support the notion that the late postpartum period is particularly high risk for maternal death secondary to self-harm [13]; this strengthens the importance of studying preventative interventions, including pharmacotherapy, for risk of SHI during the entire postpartum year.

Antidepressants, in particular selective serotonin reuptake inhibitors (SSRIs), would be expected to reduce thoughts of self-harm by serotonin regulation [14,15], but evidence for their effect on the repetition of SHI in non-pregnant subjects remains inconclusive [16]. There have also been concerns and vivid debates on the increased risk of suicidality following antidepressant use among adolescents and younger adults [17]. Whether antidepressants may, depending on their dose, patient’s vulnerability and age, worsen SHI remains unresolved [17,18,19,20]. When it comes to the population of perinatal women, research on antidepressant effectiveness is very scant [21].

Albeit with a certain degree of uncertainty, Swanson et al. [22] found no association between antidepressant continuation in pregnancy and self-harm later in gestation, relative to discontinuation (Odds Ratio (OR): 1.1, 95% Confidence Interval (CI): 0.5–2.4). Yet, the preventative association of antidepressant treatment in pregnancy on postpartum SHI has never been explored before, despite the call for more research on this topic [7,10,23]. Addressing this knowledge gap is important, as it can aid clinicians and women in weighing possible risks of antidepressant medication in pregnancy against their benefits to mental health, and thereby facilitate a more informed decision-making about pharmacotherapy in pregnancy in women with ongoing psychiatric illness [7,24].

This study sought to estimate the association between antidepressant treatment during pregnancy and postpartum SHI based on item 10 of the Edinburgh postnatal depression scale (EPDS), in a population of women with preexisting or new onset depression and/or anxiety during pregnancy. The postpartum period covered the time from five weeks to one year post-birth.

## 2. Materials and Methods

This is a sub-study of the ‘Multinational Medication Use in Pregnancy Study [25], a multinational, cross-sectional, web-based investigation to examine patterns and correlates of medication use in pregnancy. Data were collected via a self-administered anonymous questionnaire (www.questback.com) in 18 countries. Women located in Europe, North and South America and Australia who were pregnant or who had given birth less than a year ago were eligible to participate. The study was advertised on 2–3 pregnancy-related websites in each country, pregnancy forums and social media, and was open to the public for two months between October 2011 and February 2012 in each participating country. The recruitment national websites were selected for having the greatest number of daily users. The full questionnaire and further details about recruitment and the tools have been previously published [25].

Women from countries outside of those advertised and if fewer than 100 women responded were excluded from the analysis. To limit the risk of outcome misclassification due to ‘baby-blues’, which are common in the first few weeks postpartum, we excluded women with newborns between 0–4 weeks old at the time of questionnaire completion from the main analyses. Those who did not respond to the outcome variable, were pregnant or did not report a long-term psychiatric disorder in pregnancy, were excluded. The final study population included 187 postpartum women who reported having preexisting or new onset psychiatric disorders, mainly depression and/or anxiety during pregnancy (*n* = 182); very few women reported other conditions including two-sided personality, panic (*n* = 1), bipolar (*n* = 1) or obsessive-compulsive disorder (*n* = 1). This is to ensure that all women in the sample had a non-zero probability of being exposed to antidepressants during their pregnancy, and allow fairer comparison between medicated and non-medicated with antidepressants [26].

### 2.1. Postpartum Self-Harm Ideation

The outcome variable is reported thoughts of self-harm within one year after childbirth, as measured by item 10 of the EPDS. We examined SHI from five weeks post-birth to end of first postpartum year as this entire period is considered to be high risk for maternal self-harm and suicidality [13,27]. Women reported their SHI by answering the EPDS item 10, at the time of questionnaire completion, from early to late postpartum. The EPDS is widely used internationally to screen for depressive symptoms in prenatal and postpartum women [28,29]. The EPDS was provided in the mother-tongue of the country it was being administered in and validity across countries has been explored (see Appendix A) [29]. The women were asked ten questions related to how they were feeling in the last seven days concerning symptoms of guilt, sadness, anhedonia, and suicidal ideation. This latter construct, i.e., the EPDS item 10 reads “*The thought of harming myself has occurred to me*”. Available responses are “Never”, “Hardly ever”, “Sometimes”, and “Yes, quite often”, which are assigned 0, 1, 2, and 3 points, respectively. The EPDS item 10 has been used in prior research to measure self-harm thoughts [11,23,30,31]. Women were considered to have frequent SHI if they reported SHI frequency “quite often” or “sometimes” (i.e., with score >1 on item 10 of the EPDS). This cut-off has been previously used in the literature [31]. No SHI was defined as answering “Hardly ever” or “Never” to item 10 of the EPDS in the main analysis. To further disentangle total absence of SHI from hardly ever, we additionally compared women with SHI to each of these control groups. We also compared more sporadic SHI, defined as “Hardly ever”, to “Never”, as “hardly ever” does not mean the absolute absence of SHI. The other EPDS items (1–9) were summed (score range: 0–27) to quantify concurrent symptoms of depression and anxiety at the time of reporting on SHI.

### 2.2. Self-Reported Depression and Anxiety, and Antidepressant Treatment

Participating women were invited to respond to a set of questions regarding the presence of long-term and/or chronic diseases during pregnancy, including depression and anxiety. A free-text field was also available, where any other condition not previously listed could be specified. Women were not required to indicate whether the diseases were new onset or pre-existing from before the pregnancy. Because depression is highly comorbid with anxiety, and antidepressant is indicated for the treatment of both disorders [32], women who reported having long-term depression and/or anxiety met the study inclusion criteria. In addition, we included isolated cases of women who reported other psychiatric disorder (i.e., bipolar, panic or personality disorders) during pregnancy in the free-text field. The rationale for the inclusion is that antidepressants may be used for these illnesses [32].

Women could then report if they had used medications for each long-term/chronic disease during their pregnancy as free-text entry, along with the timing of usage (pregnancy weeks 0–12, 13–24, 25–delivery). We classified reported medications using the Anatomical Therapeutic Chemical (ATC) Classification system into the following categories: SSRI (ATC code N06AB), serotonin and norepinephrine reuptake inhibitor (SNRI) (ATC codes N06AX, venlafaxine and duloxetine), tricyclic antidepressants (TCA) (ATC code N06AA) and unspecified antidepressant (ATC code N06A-). As a proxy of the duration of treatment, we counted how many trimester intervals were checked in the questionnaire. Antidepressant use ever in pregnancy and duration in trimesters constituted the exposure variables. Because few women took antidepressants for two trimesters, we examined SHI in relation to one or three trimesters only. The study did not collect data about medication use after childbirth.

### 2.3. Covariates

Baseline characteristics included measures related to the pregnancy, such as the use of folate before and/or during pregnancy, if the pregnancy was planned, having previous children, child age (in weeks) at time of questionnaire completion, and healthcare contact due to infertility. Maternal sociodemographic and life-style factors included age, working status at conception, marital status, educational level, mother tongue, smoking habit during pregnancy, alcohol consumption after awareness of pregnancy, and region of residency. For the latter variable, participating countries were grouped into the regions (1) Western Europe (including France, Italy, Switzerland, and the UK); (2) Northern Europe (including Finland, Norway and Sweden), (3) Eastern Europe (including Croatia, Poland, Russia, Serbia and Slovenia); (4) North America (including the USA and Canada) and (5) Australia. 

The survey included questions about the personality trait neuroticism, which was measured using eight items of the Big Five Inventory (BFI) [33]. We included this factor in the analyses as neuroticism is highly correlated with psychiatric disorders [34]. Health-related measures included the number of reported psychiatric disorders in pregnancy, as well as depressive symptoms based on items 1–9 of the EPDS, and having other chronic/long-term disorders in pregnancy (i.e., asthma, allergy, hypothyroidism, cardiovascular or rheumatic disease, diabetes type I or II) as self-reported within the list of chronic disorders. Women were additionally invited to report whether they had experienced nausea or sleeping problems during pregnancy, within a list of common acute pregnancy-related illnesses. Use of other medication in pregnancy was defined according to the ATC system and included benzodiazepines and z-hypnotics (ATC codes N05BA, N05CD, N05CF), and antipsychotics (ATC code N05A).

### 2.4. Data Analysis

To make results more generalizable, we generated sampling weights according to publicly available data on education levels by maternal age in each participating country [35,36,37,38], as age and education are important determinants of study response. Each woman was assigned a weight, obtained by dividing the proportion in the birthing population by the corresponding proportion in our sample, in each age-by-education stratum [39]. Women under-represented in our sample were assigned a weight greater than one, while those over-represented received a weight smaller than one. Descriptive statistics were then conducted before (crude) and after (weighted, by sampling weight) application of the survey weight to the data.

To estimate the association between past antidepressant treatment during pregnancy and postpartum SHI, we applied inverse probability of treatment weighting (IPTW), using the propensity score with survey data, as described by DuGoff et al. [40] Propensity scores were used to balance the baseline characteristics between women medicated with antidepressants in pregnancy and the non-medicated. Propensity scores, i.e., the probability of “exposure” to an antidepressant, were generated using logistic regression with exposure (i) ever in pregnancy, (ii) in three trimesters, or (iii) in one trimester as the dependent variables, given a set of maternal covariates (cf. Appendix A) and the survey weight. We then derived and normalized the IPTW. The aim was to reduce differences in the distribution of covariates between the exposure groups to ≤0.1 standardized mean difference [41]. The final weight applied to the regression outcome models was comprised of the sampling weight multiplied by the IPTW. We then fit crude and weighted modified Poisson regression models within the generalized linear model framework, to compute risk ratio (RR) with 95% Confidence Interval (CI) [42]. Risk ratio estimation was considered preferable over that of ORs as the prevalence of SHI was common in our sample. Data are reported as crude and weighted RR with 95% CI, as well as weighted point prevalence with 95% CI. All analyses were conducted in Stata MP version 15/16 (StataCorp LLC College Station, Texas, USA).

Less than 4% of the women had missing values in at least one of the covariates (see footnote Table 1). Under the assumption that data were missing at random, we imputed incomplete data via multiple imputation with chained equation (five replications) [43].

### 2.5. Sensitivity Analyses

To examine the robustness of our results, we conducted a set of sensitivity analyses, as described in the Appendix A. To account for country variation, we conducted a weighted multilevel mixed-effect regression model with inclusion of a random effect for country of residence. Because it is still debated whether antidepressants may increase the risk of SHI in younger adults individuals [17,18,19], we tested for an interaction term between antidepressant treatment in pregnancy and maternal age, as continuous variable and categorized as 17–24 years, 25–30, and over 30 years. To further elucidate this latter point, we used as positive control outcome the postpartum EPDS score on items 1–9, as we would expect reduced EPDS symptoms among medicated relative to non-medicated [44], via crude and weighted linear regression models. Because SHI was measured at different postpartum times, which is from childbirth to child age one year, we tested the interaction term between timing of SHI report and antidepressant use in pregnancy in the weighted models.

### 2.6. Ethics Approval and Informed Consent

Informed consent was given by the participants by ticking the answer “yes” to the question “Are you willing to participate in the study?” The Regional Ethics Committee in Norway, region South-East, granted an ethical approval exemption for this study because of anonymity. Ethical approval or study notification to the relevant national Ethics Boards was achieved in the UK and Italy as required by the national legislation. All data were handled and stored anonymously.

## 3. Results

The final study population included 187 postpartum women who had reported a long-term psychiatric disorder during pregnancy. The selection criteria of the final study population is displayed in Figure 1. Most women (57.2%) had a child of age 8–12 months at the time of study participation. The most commonly reported psychiatric disorders were depression and anxiety, either alone or comorbid (*n* = 182). Five women reported other conditions including two-sided personality (*n* = 2), panic (*n* = 1), bipolar (*n* = 1) or obsessive-compulsive disorder (*n* = 1).

Based on of maternal self-report, 88 (47.1%) women did not take antidepressants during pregnancy, whilst 99 (52.9%) did so. The unweighted baseline characteristic of the study population, overall and by antidepressant treatment during pregnancy, are presented in Table 1. There was no major difference between women medicated with antidepressants and non-medicated in relation to time of questionnaire response. Those medicated were more often older, from countries other than Eastern Europe, employed and with higher educational attainment than non-medicated. However, non-medicated women consumed alcohol more often than medicated and had greater depressive symptoms based on items 1–9 of the EPDS.

As shown in Table 2, SSRIs were the most commonly taken antidepressants, followed by SNRIs. Sertraline (*n* = 25, 13.4%), fluoxetine (*n* = 21, 11.2%) and citalopram (*n* = 20, 10.7%) were the most commonly used substances (all SSRIs). Venlafaxine was the predominant SNRI (*n* = 9, 4.8%). Many women (66/99, 66.7%) reported continued antidepressant use for all three trimesters, whilst fewer took it for two (11/99, 11.1%) or a single (22/99, 22.2%) trimester. This single trimester of antidepressant use coincided in most cases (19/22, 86.4%) with the first trimester of pregnancy. There were five women using combination therapy of two antidepressant classes.

Figure 2 and Table 2 show the weighted proportions of SHI frequency in women medicated and non-medicated with antidepressants, ever in pregnancy as well as by the number of trimesters and classes. Women who were medicated with antidepressants during pregnancy reported more often SHI in the postpartum period (22.0% vs. 15.2%) relative to non-medicated; total absence of SHI postpartum was greater in medicated women than in non-medicated (71.7% vs. 62.3%). Women who took antidepressants for a single trimester had a greater SHI postpartum than those with continued use in all three trimesters, although the difference between groups is not statistically significant. The survey weighting adjustment yielded a change in the proportions of SHI frequency in the range 0.7–4.3% (see Appendix A).

Table 3 shows crude, survey-weighted, and fully weighted RR for SHI by past antidepressant treatment in pregnancy. After controlling for confounding via weighting, the rate of frequent SHI was 90% higher among women ever treated with antidepressant in pregnancy compared to non-medicated, albeit the 95% CI included the null effect. This elevated proportion was driven by antidepressant use in one trimester, whilst the RR for continued exposure in three trimesters was close to 1. Comparison of frequent SHI with never SHI, attenuated the magnitude of the positive association measures. Women medicated with antidepressants in pregnancy had a lower rate of reporting sporadic SHI than the non-medicated counterpart (weighted RR: 0.43, 95% CI: 0.17–1.09), but the evidence for this association was weak (*p* = 0.075) after weighting. Balance between covariates before and after application of the IPTW is depicted in Appendix A.

Results of the various sensitivity analyses are described in the Appendix A. There was no statistically significant interaction between antidepressant use in pregnancy and maternal age in all outcome models, and likewise with time postpartum when SHI was reported. In the positive control outcome analysis, women medicated for three trimesters had lower postpartum EPDS score on items 1–9 (mean difference, −2.54) relative to non-medicated women, after controlling for confounding via weighting. Women medicated in a single trimester had greater EPDS score, but the 95% CI was broad and included the null (see Appendix A).

## 4. Discussion

This study reports new knowledge on the association between antidepressant treatment in pregnancy and self-harm thoughts in the postpartum year. In a population of women who reported depression or anxiety during gestation, there was no evidence that antidepressant treatment in pregnancy was associated with a lower occurrence of frequent SHI in the postpartum year. A reduced rate appeared for more sporadic SHI, albeit the evidence was weak, and the clinical relevance of this association seems limited. Although based on few cases among the exposed, the rate of more frequent SHI was elevated following antidepressant treatment for a single trimester, whilst that was comparable in women medicated with antidepressants across all three trimesters and non-medicated.

Concerns about antidepressants and the composite outcome of suicidality, including attempted or completed suicide, suicidal behavior and self-harm, have been the topic of vivid debate in the last three decades [17]. There is now convincing evidence for an increasing risk of suicidality with antidepressants at treatment initiation and in younger adults (18–24 years). However, no risk difference is found in the non-pregnant population at age 25–30 years [45,46]. Our association between antidepressants in all trimesters and postpartum SHI was close to the null, which broadly aligns with findings from other studies [45,46]. This result may also be explained by the fact that antidepressant treatment was most likely not initiated in pregnancy, but already ongoing prior to conception [2]. The study could not verify any age-specific associations, possibly because there were few women in the younger age band of 18–24 years, and residual confounding by maternal age cannot be ruled out. In addition, SHI was self-reported, which may have minimized the risk of detection bias of suicidal thoughts in younger women.

The preventative association of antidepressant in pregnancy on more sporadic SHI, but not on more frequent SHI in the postpartum year, is surprising and somewhat in contrast with prior knowledge on the effectiveness of antidepressant in more severe depression [21,47]. Even though the evidence for the association was weak and its clinical relevance seems limited, this contrasting finding raises concerns about residual confounding by severity of maternal mental illness, and about the reliability of hardly ever self-harm classification on the EPDS item 10 [11,31]. At present, our and other available studies on antidepressants in pregnancy [44] do not document a benefit of antidepressant on self-harm reduction seen in observational studies among non-pregnant adults [15,48]. Because the interplay between the perinatal hormonal fluctuations and the pharmacological action of antidepressants remains elusive, we cannot exclude a more definite benefit of this drug treatment in pregnancy on other core symptoms of postpartum depression such as anhedonia or sadness [44,49].

One key finding is that postpartum SHI was more elevated in women who had been medicated with antidepressants for a single trimester relative to non-medicated ones, whilst for antidepressants in all three trimesters the SHI rate was comparable to the no medication group. It is possible that women medicated in a single trimester, which coincided with the first one in most cases, discontinued their antidepressant upon recognition of the pregnancy. It is known that pregnancy is a well-established driver for antidepressant discontinuation, irrespective of underlying severity of maternal mental illness [27,50]. Antidepressant discontinuation poses a 60% increased risk for suicide attempt in non-pregnant subjects, although its effect on self-harm is unclear [22,51]. Our effect estimates for single trimester exposure coupled to the greater EPDS score on items 1–9 in these women, provide some insights into the importance of continuing a needed treatment with antidepressant in women with mental illness to safeguard maternal mental health and the wellbeing of the child.

The heightened proportion of more frequent SHI in women taking SNRIs, compared to that among SSRI users, was based on very few women. A methodologically sound study showed no difference in the rate of deliberate self-harm in individuals initiating either an SSRI or an SNRI (2.5 vs 2.8 per 1000 person-years) [19]. SSRIs represent the preferred pharmacotherapy choice in pregnancy, and SNRI are often reserved to more severe cases of depression [32]. Thus, our elevated rate of SHI among SNRI treated women may be confounded by maternal mental illness severity. Future longitudinal studies with greater statistical power will be crucial in determining the real-world effectiveness of SSRI and SNRI treatments in perinatal women, and whether this pharmacotherapy may be of benefit in reducing the risk of SHI in the postpartum year. More research is necessary to rule out possible differential risks for SHI following antidepressant initiation or discontinuation according to maternal age. How antidepressant treatment during pregnancy effects women in their postpartum period is an important question to answer to empower women to make evidence informed decisions about their mental health.

### Strengths and Limitations

One strength of the study is the use of a measure of self-harm, based on item 10 of the EPDS. Although this specific item has not been explicitly validated against clinical interview, the EPDS scale as a whole has been validated extensively and in various subgroups of women; also, the EPDS is designed to screen for perinatal mood specifically and has satisfactory (0.88) reliability [11,29]. Due to the sensitive nature of the variables of interest, having the survey be anonymous from the beginning is also a strength. Data collection was conducted uniformly in all participating countries via utilization of an anonymous electronic questionnaire, and we corrected our association and descriptive estimates by survey weight adjustment, allowing the findings to be more generalizable in terms of age and education. The IPTW approach minimized the difference in baseline characteristics between medicated and non-medicated women. Restriction of the sample to women with psychiatric disorders in pregnancy allowed fairer comparisons between antidepressant exposed and unexposed [26]. We also imputed missing data on covariates, which is a methodological advantage [43].

One important limitation is the cross-sectional design of the study and the lack of the temporal component. However, treatment with antidepressants was reported retrospectively by mothers, which make their associations with postpartum SHI response valid. The risk of differential outcome misclassification is unlikely in this setting, as this risk is more prominent when studying negative child outcomes following medication in-utero exposures [52]. We measured SHI only once in the postpartum year, and although the EPDS has been validated across countries [29], how women interpret frequencies belonging to hardly ever to sometimes or often, is subjective. Yet, multiple studies have used same EPDS item to measure SHI [11,23,30,31]. The psychiatric disorders and antidepressant use were self-reported by the participants, and thus dependent on the accuracy of the woman’s reporting. However, the psychiatric correlates of our population are suggestive of high morbidity, and women were specifically asked to indicate whether they had long-term depression or anxiety. We had no data on the time of onset of the psychiatric disorder, i.e., before or during pregnancy, or SHI status in pregnancy. The study did not measure use of antidepressants after pregnancy, or other postpartum occurrences such as domestic violence, that could be risk factors for SHI; however, the study aimed to estimate the total effect of antidepressant in pregnancy on postpartum SHI, and not its direct effect independent of antidepressant treatment after childbirth. Also, factors occurring after the end of exposure window, are by definition, not confounding factors for the associations investigated here [53]. Both dose and exact timing of antidepressant use are unknown, and we used the number of trimesters as a proxy of treatment duration. The underlying reason for antidepressant use in a single trimester versus three trimesters (i.e., self-initiated by the woman or following advice by the clinician) is unknown. Unmeasured confounding by suicidality history, life-time abuse or partner violence cannot be ruled out. The small sample size limited the statistical power of specific analyses, and we were unable to detect small effect sizes. Moreover, the imprecision of the effect estimates, with overlapping confidence intervals, make the interpretation of trimester specific findings challenging.

The questionnaire was only available through internet websites, which did not permit calculation of a conventional response rate and a selection bias cannot be ruled out. However, recent epidemiological studies indicate reasonable validity of web-based recruitment methods [54,55] and the internet penetration rate, either in households or at work, is relatively high among women of childbearing age [56]. We have previously assessed the study’s external validity on an individual country and found that on average, the women in the study had higher education and were slightly more often primiparous than the general birthing populations in the various countries [25]. Although we made our study more generalizable in terms of age and education via survey weighting, selection bias due to access to internet cannot be excluded, and we cannot exclude the possibility that the women who decided to participate in the study differed from the general birthing population with depressive or anxiety disorders in ways that our analysis could not control for.

## 5. Conclusions

In a population of women with antenatal depression/anxiety, there was no preventative association between past antidepressant treatment in pregnancy and reporting frequent SHI in the postpartum year. Women medicated in a single trimester reported postpartum SHI more often than non-medicated comparators. Our findings are only a first step in providing evidence to inform psychiatric disorder treatment decisions for pregnant women, and future investigations are encouraged to determine the real-world effectiveness of antidepressant in perinatal psychiatry. Women should be empowered to develop an evidence-based understanding, not solely of the potential risks, but also of the benefits of antidepressant treatment in pregnancy, in order to optimize maternal-child health.

## Figures and Tables

**Figure 1 ijerph-18-00046-f001:**
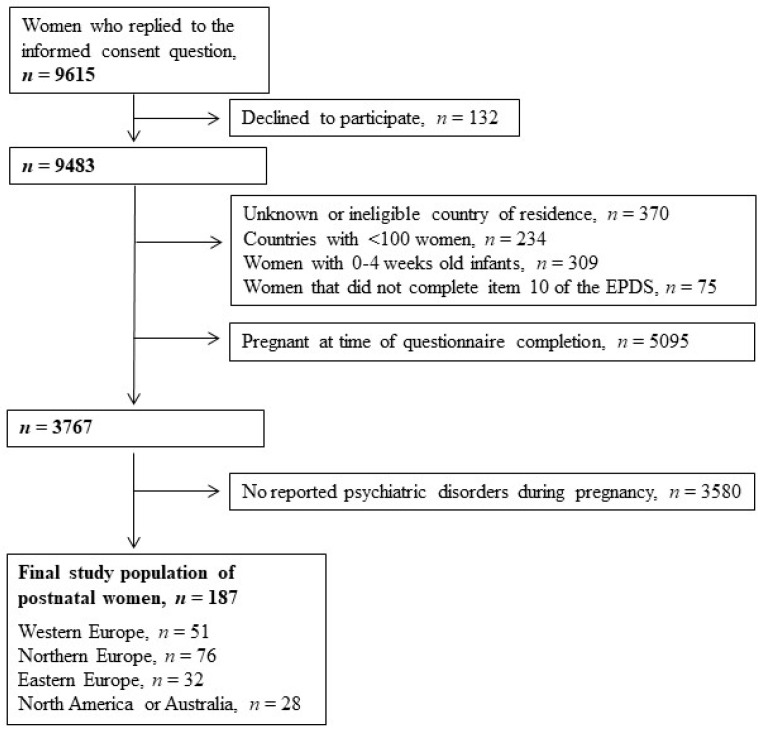
Data flow to achieve the final study population. Abbreviations: EPDS = Edinburgh Postnatal Depression Scale. Psychiatric disorders include depression, anxiety and other psychiatric disorders (i.e., bipolar, panic and two-sided personality disorders).

**Figure 2 ijerph-18-00046-f002:**
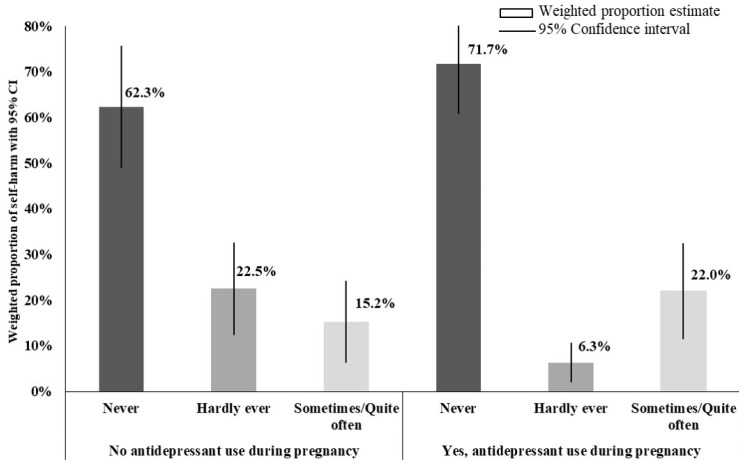
Weighted proportion of frequency of self-harm ideation by antidepressant use among 187 women with depression/anxiety during pregnancy. Magnitude of proportions may not directly align with number of women in the various categories because proportions are weighted via survey weighting method. Antidepressants included all selective serotonin re-uptake inhibitors (SSRIs, ATC code N06AB), serotonin and norepinephrine reuptake inhibitor (SNRI, ATC codes N06AX, venlafaxine and duloxetine), tricyclics (TCA, ATC code N06AA), and unspecified antidepressant (ATC code N06A-).

**Table 1 ijerph-18-00046-t001:** Maternal sociodemographic, lifestyle and health-related characteristics of the study population by antidepressant treatment during pregnancy ^1^.

Sociodemographic, Life-Style and Health Characteristics	Study Population	Antidepressant Treatment during Pregnancy	
	Total, *n* = 187	No, *n* = 88	Yes, *n* = 99
	*n* (%)	*n* (%)	*n* (%)
*Pregnancy characteristics*			
Weeks since childbirth			
5–16	42 (22.5)	19 (21.6)	23 (23.2)
17–28	38 (20.3)	17 (19.3)	21 (21.2)
29+	107 (57.2)	52 (59.1)	55 (55.6)
Nulliparous (yes)	83 (44.4)	40 (45.5)	43 (43.4)
No perinatal use of folate	11 (5.9)	4 (4.6)	7 (7.1)
Unplanned pregnancy (yes)	33 (17.7)	15 (17.1)	18 (18.2)
*Sociodemographic and life-style characteristics*			
Region of residency ^2^			
Western Europe	51 (27.3)	21 (23.9)	30 (30.3)
Northern Europe	76 (40.6)	27 (30.7)	49 (49.5)
Eastern Europe	32 (17.1)	28 (31.8)	4 (4.0)
North America	20 (10.7)	10 (11.4)	10 (10.1)
Australia	8 (4.3)	2 (2.3)	6 (6.1)
Maternal age (years)			
17–24	38 (20.3)	28 (31.8)	10 (10.1)
25–30	66 (35.3)	30 (34.1)	36 (36.4)
>30	83 (44.4)	30 (34.1)	53 (53.5)
Married or cohabitating (yes)	163 (87.2)	74 (84.1)	89 (89.9)
Working status at conception			
Employed	99 (52.9)	43 (48.9)	56 (56.6)
Student	30 (16.0)	18 (20.5)	12 (12.1)
Homemaker	30 (16.0)	15 (17.1)	15 (15.2)
Job seeker/other	28 (15.0)	12 (13.6)	16 (16.2)
Educational Attainment			
Less than high school	26 (13.9)	15 (17.1)	11 (11.1)
High school	74 (39.6)	37 (42.1)	37 (37.4)
More than high school	87 (46.5)	36 (40.9)	51 (51.5)
Immigrant status (yes) ^3^	7 (3.7)	3 (3.4)	4 (4.0)
Smoking during pregnancy (yes)	30 (16.0)	16 (18.2)	14 (14.1)
Alcohol use during pregnancy (yes) ^4^	34 (18.2)	19 (21.6)	15 (15.2)
*Health-related factors*			
Healthcare contact due to infertility (yes)	34 (18.2)	19 (21.6)	15 (15.2)
Neurotic trait (range 8–40), mean (SD)	28.8 (5.5)	29.3 (5.4)	28.4 (5.6)
Nausea in pregnancy (yes)	147 (78.6)	67 (76.1)	80 (80.8)
Sleeping problems in pregnancy (yes)	115 (61.5)	52 (59.1)	63 (63.6)
Non-psychiatric chronic conditions (yes)	101 (54.0)	57 (64.8)	44 (44.4)
Co-medication use in pregnancy (yes)			
Antipsychotics	16 (8.6)	6 (6.8)	10 (10.1)
Benzodiazepines and z-hypnotics	27 (14.4)	9 (10.2)	18 (18.2)
Number of co-morbid psychiatric disorders			
1	103 (55.1)	50 (56.8)	53 (53.5)
2	80 (42.8)	37 (42.1)	43 (43.4)
3	4 (2.1)	1 (1.1)	3 (3.0)
EPDS score (items 1–9; range 0–27), mean (SD)	11.6 (5.9)	12.7 (5.8)	10.6 (5.0)

^1^ The table show crude, non-weighted proportions. Numbers may not add up to total due to missing values (<4%) for 7 women on neuroticism traits (*n* = 5), alcohol use (*n* = 2), immigrant status (*n* = 1) and unplanned pregnancy (*n* = 1). Antidepressants included all selective serotonin re-uptake inhibitors (SSRIs, ATC code N06AB), serotonin and norepinephrine reuptake inhibitor (SNRI, ATC codes N06AX, venlafaxine and duloxetine), tricyclics (TCA, ATC code N06AA), and unspecified antidepressant (ATC code N06A-). ^2^ Western Europe includes France, Italy, Switzerland, and the United Kingdom; Northern Europe includes Finland, Norway and Sweden; Eastern Europe includes Croatia, Poland, Russia, Serbia and Slovenia; North America includes USA and Canada. ^3^ Women having the first language different from the official main language in the country of residency. ^4^ Indicates alcohol use after awareness of pregnancy.

**Table 2 ijerph-18-00046-t002:** Weighted proportion of frequency of postnatal self-harm according to past antidepressant treatment in pregnancy, by duration and class (*n* = 187) ^1^.

		Frequency of Postnatal Self-Harm		
		Sometimes/ Quite Often	Hardly Ever	Never
	*n*	% (95% CI)	% (95% CI)	% (95% CI)
Non-medicated in pregnancy	88	15.2 (6.3–24.2)	22.5 (12.4–32.6)	62.3 (48.9–75.6)
Any antidepressant in 1 trimester	22	36.3 (8.6–64.0)	-	63.7 (36.0–91.4)
Any antidepressant in 2 trimesters	11	10.0 (0.08–28.8)	18.1 (1.0–41.3)	71.9 (44.3–99.4)
Any antidepressant in 3 trimesters	66	18.0 (7.5–28.5)	7.1 (1.5–12.7)	74.9 (63.4–86.4)
SSRI antidepressant, ever in pregnancy	85	11.3 (3.6–18.9)	6.5 (1.8–11.3)	82.2 (73.5–91.0)
SNRI antidepressant, ever in pregnancy	14	70.1 (44.9–95.4)	5.1 (1.0–15.4)	24.7 (1.7–47.8)

Abbreviations: SNRI, serotonin and norepinephrine reuptake inhibitor; SSRI, selective serotonin reuptake inhibitor. ^1^ There were five cases of combination therapy with two medication classifications. Only exposure data for at least five women are reported: 4 women reported used of tricyclic antidepressants, and 1 woman reported used of an unspecified antidepressant. Antidepressants included all selective serotonin re-uptake inhibitors (SSRIs, ATC code N06AB), serotonin and norepinephrine reuptake inhibitor (SNRI, ATC codes N06AX, venlafaxine and duloxetine), tricyclics (TCA, ATC code N06AA), and unspecified antidepressant (ATC code N06A-).

**Table 3 ijerph-18-00046-t003:** Association between antidepressant treatment during pregnancy and postnatal SHI ^1^.

	*n*	*n* (%) with SHI	Crude RR (95% CI)	Survey Weighted, ^2^ RR (95% CI)	Fully Weighted, ^3^ RR (95% CI)
*Frequent SHI vs never/hardly ever SHI*					
Non-medicated in pregnancy	88	14 (15.9)	Reference	Reference	Reference
Antidepressants, ever in pregnancy	99	18 (18.2)	1.14 (0.60–2.17)	1.44 (0.68–3.08)	1.90 (0.79–4.56)
Antidepressants for 3 trimesters	66	11 (16.7)	1.05 (0.50–2.18)	1.18 (0.52–2.71)	0.95 (0.38–2.39)
Antidepressants for 1 trimester	22	6 (27.3)	1.71 (0.73–4.0)	2.38 (0.90–6.29)	1.85 (0.60–6.04)
*Frequent SHI vs never SHI* ^4^					
Non-medicated in pregnancy	65	14 (21.5)	Reference	Reference	Reference
Antidepressants, ever in pregnancy	90	18 (20.0)	0.93 (0.50–1.74)	1.19 (0.55–2.57)	1.66 (0.68–4.02)
Antidepressants for 3 trimesters	59	11 (18.6)	0.87 (0.42–1.78)	0.99 (0.43–2.29)	0.85 (0.34–2.17)
Antidepressants for 1 trimester	22	6 (27.3)	1.27 (0.55–2.94)	1.85 (0.69–4.93)	1.49 (0.45–4.94)
*Sporadic SHI vs never” SHI* ^5^					
Non-medicated in pregnancy	74	23 (31.1)	Reference	Reference	Reference
Antidepressants, ever in pregnancy	81	9 (11.1)	0.36 (0.18–0.73) *	0.31 (0.13–0.70) *	0.43 (0.17–1.09) ^§^
Antidepressants for 3 trimesters	55	7 (12.7)	0.41 (0.19–0.89) ^†^	0.33 (0.13–0.81) ^¶^	0.57 (0.18–1.80)

Abbreviations: SHI = self-harm ideation; RR = Risk Ratio; CI = Confidence Interval. ^1^ Antidepressants included all selective serotonin re-uptake inhibitors (SSRIs, ATC code N06AB), serotonin and norepinephrine reuptake inhibitor (SNRI, ATC codes N06AX, venlafaxine and duloxetine), tricyclics (TCA, ATC code N06AA), and unspecified antidepressant (ATC code N06A-).^2^ Weighted only by the survey weight, accounting for maternal age and education. ^3^ Weighted with the composite weight, constructed by multiplying the survey weight with the stabilized inverse probability of treatment weighting using maternal baseline covariates. ^4^ Observations reporting “hardly ever” self-harm thoughts are excluded in this comparison. ^5^ Observations reporting “quite often/sometimes” self-harm thoughts are excluded in this comparison. There was no woman on antidepressant use for one trimester with the outcome, and so this analysis was not conducted. * *p*-value = 0.005; ^§^
*p*-value = 0.075; ^†^
*p*-value = 0.025; ^¶^
*p*-value = 0.017.

## Data Availability

The data presented in this study are available on request from the corresponding author.

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
