# Peer review of "Association between Antidepressant Treatment during Pregnancy and Postpartum Self-Harm Ideation in Women with Psychiatric Disorders: A Cross-Sectional, Multinational Study"

_ijerph, 2020, doi:10.3390/ijerph18010046_

Round 1

Reviewer 1 Report

Intruiging data, but does this reflect more ‘severe’ cases, or poorly treated (cfr single trimester). Another way to address these data is that more severe cases treated with antidepressant had similar outcome after delivery ? The link to the similar discussion in teenagers is wel taken.

Were the depression or anxiety specific to pregnancy, or were cases also included if pregnancy occur with a history or active depression or anxiety. Do I understand it correct that only ‘naïve’ cases were included ? if so, this should perhaps also be mentioned or reflected in the abstract and/or title ?

Is it correct that postpartum was defined as the first year after delivery (as different definitions exist on this). this is pragmatic approach, but should perhaps also be reflected in the abstract ? 

There is a textual issue on line 110, please check

Similar comment on line 127 (why bold ?).

Is the absence of data on postpartum drug use not a major limitation ?

Based on the study design, can there a bias in recall (cfr table 1, weeks since childbirth)?

If women took antidepressants in the first trimester, does this reflect a portion of cases that stopped treatment because of pregnancy ?

I value the work, but based on this design, suicide events can obviously not be collected.

Are references 25 and 58 the same ?

Author Response

We thank the Editor and the Reviewers for the opportunity to revise our manuscript and for the valuable feedback provided. Please find below our replies to each individual comment.

Reviewer 1

Comment 1: Intruiging data, but does this reflect more ‘severe’ cases, or poorly treated (cfr single trimester). Another way to address these data is that more severe cases treated with antidepressant had similar outcome after delivery ? The link to the similar discussion in teenagers is wel taken.

Reply 1: Thank you for these reflections. We agree that interpretation of results warrants careful reflection. We do believe that the results on difference in the Edinburgh Postnatal Depression Scale (EPDS) scores between first trimester continuers and discontinuers supports the interpretation that discontinuation in first trimester results may lead to more unfavorable postnatal outcomes than better-treated disease (see Table S3). Importantly, women with antidepressant use in three trimesters had lower depressive symptoms scores than non-medicated, while this was not the case for women medicated in a single trimester. This point is discussed in lines 327-337 of the manuscript.

To address the reviewers comment, we have added the following sentences in the limitation section: 

Lines 397-398 of the revised manuscript: “The underlying reason for antidepressant use in a single trimester versus three trimesters (i.e. self-initiated by the woman or following advice by the clinician) is unknown.”

Lines 401-402 of the revised manuscript: “Moreover, the imprecision of the effect estimates, with overlapping confidence intervals, make the interpretation of trimester specific findings challenging.” 

Comment 2: Were the depression or anxiety specific to pregnancy, or were cases also included if pregnancy occur with a history or active depression or anxiety. Do I understand it correct that only ‘naïve’ cases were included ? if so, this should perhaps also be mentioned or reflected in the abstract and/or title ?

Reply 2: Thank you for raising this point. The study included women reporting longer-term depression or anxiety in pregnancy, which may include new onset cases in pregnancy and pre-existing cases. We have now made this clearer throughout the manuscript, including the abstract, by rephrasing relevant text passages to “preexisting or new onset depression and/or anxiety during pregnancy”. In the Methods sub-section “Self-reported depression and anxiety, and antidepressant treatment” we have now clarified this, which reads: “The questionnaire did not inquiry women whether the diseases were pre-existing from before the pregnancy or new onset conditions”. This issue was also discussed as a limitation of the study, see lines 389-390.

Comment 3: Is it correct that postpartum was defined as the first year after delivery (as different definitions exist on this). this is pragmatic approach, but should perhaps also be reflected in the abstract ? 

Reply 3: Yes, that is correct. We have now made this clearer in the abstract and throughout the manuscript. However – as specified in the Methods – we excluded from the sample women with newborns between 0-4 weeks old at the time of questionnaire completion from the main analyses, to limit risk of misclassification due to ‘baby-blues’, which are common in the first few weeks postpartum.

Comment 4: There is a textual issue on line 110, please check. Similar comment on line 127 (why bold ?).

Reply 4: We have now fixed the issue in line 127, but could not identify any bold in line 110. Any remaining text issue will be solved when doing the proof of the article.

Comment 5: Is the absence of data on postpartum drug use not a major limitation ?

Reply 5: We stated this as limitation of the study in lines 390-391 of the manuscript. However, as discussed, we were interested in the total preventative association of antenatal antidepressant use on postnatal SHI. Antidepressant use postpartum would act as mediator on this path, not as confounder.

Comment 6: Based on the study design, can there a bias in recall (cfr table 1, weeks since childbirth)?

Reply 6: Thank you for this comment. In the Limitation section, we stated: “The risk of differential outcome misclassification is unlikely in this setting, as this risk is more prominent when studying negative child outcomes following medication in-utero exposures”. In this sentence we recognize the recall on past antidepressant use in women at different postpartum times; however, we feel this is not differential and thus it is unlikely that it will bias our association measures. To support this, we conducted a sensitivity analysis and tested the statistical significance of the interaction term between antidepressant treatment and child age at the time of questionnaire completion (see supplementary material). We did not identify any interaction between response time and antidepressant use.

Later in the Limitation, we also stated that “The psychiatric disorders and antidepressant use were self-reported by the participants, and thus dependent on the accuracy of the woman’s reporting”.

Comment 7: If women took antidepressants in the first trimester, does this reflect a portion of cases that stopped treatment because of pregnancy ?

Reply 7: Thank you for raising this comment. When interpreting our results (see lines 320-327), we proposed the possibility that women medicated in a single trimester, which coincided with the first one in most cases, discontinued their antidepressant upon recognition of the pregnancy. This is supported by the fact that pregnancy is a well-established driver for antidepressant discontinuation. However, our study did not collect information about the underlying reason for discontinuation, ie if that was following advice by the clinician or woman’s self-initiative. We have now added this point as additional limitation, which reads: “The underlying reason for antidepressant use in a single trimester versus three trimesters (i.e. self-initiated by the woman or following advice by the clinician) is unknown.”

Comment 8: I value the work, but based on this design, suicide events can obviously not be collected.

Reply 8: Thank you for the positive feedback. The study was anonymous, and so we do not have the possibility to link these data to cause of death registries in the various countries.

Comment 9: Are references 25 and 58 the same ?

Reply 9: Yes, thank you for noting this error. We have now fixed the issue.

Reviewer 2 Report

The article is interesting, the topic is actual. This research has practical significance, because maternal health is significant value. As shown, during pregnancy, 2-6% of women take antidepressants to treat perinatal psychiatric disorders, mainly depression and anxiety. Women with an antenatal mental illness are at high risk for sustained illness during the perinatal period and/or for a relapse up to one year postpartum. It is possible that antidepressant treatment in pregnancy could reduce the risk of severe mental health outcomes after childbirth, including self-harm and attempted suicide. So, the study focused on the association between antidepressant treatment during pregnancy and postpartum self-harm ideation, is very actual.

The preventative association of antidepressant treatment in pregnancy on postpartum SHI has never been explored before, despite the call for more research on this topic. Addressing this knowledge gap is important, as it can aid clinicians and women in weighing possible risks of antidepressant medication in pregnancy against their benefits to mental health, and thereby facilitate a more informed decision-making about pharmacotherapy in pregnancyin women with ongoing psychiatric illness.

However, there is one remark, on line 110 the sentence is incomplete.

Author Response

Comment 1: The article is interesting, the topic is actual. This research has practical significance, because maternal health is significant value. As shown, during pregnancy, 2-6% of women take antidepressants to treat perinatal psychiatric disorders, mainly depression and anxiety. Women with an antenatal mental illness are at high risk for sustained illness during the perinatal period and/or for a relapse up to one year postpartum. It is possible that antidepressant treatment in pregnancy could reduce the risk of severe mental health outcomes after childbirth, including self-harm and attempted suicide. So, the study focused on the association between antidepressant treatment during pregnancy and postpartum self-harm ideation, is very actual.

The preventative association of antidepressant treatment in pregnancy on postpartum SHI has never been explored before, despite the call for more research on this topic. Addressing this knowledge gap is important, as it can aid clinicians and women in weighing possible risks of antidepressant medication in pregnancy against their benefits to mental health, and thereby facilitate a more informed decision-making about pharmacotherapy in pregnancyin women with ongoing psychiatric illness.

However, there is one remark, on line 110 the sentence is incomplete.

Reply 1: Thank you for the overall positive feedback on our work, and for noticing the incomplete sentence. We have fixed the issue, and removed the incomplete text as we wrote it by mistake.

Reviewer 3 Report

The paper described the study to estimate the preventative association of antidepressants during pregnancy on postpartum self-harm ideation (SHI), using the Multinational Medication Use in Pregnancy Study. Frequency of SHI was measured via the Edinburgh Postnatal Depression Scale (EPDS). A sample of mothers who reported depression and/or anxiety during pregnancy (n=187) were included and mothers reported their antidepressant use in pregnancy retrospectively. The results of the study showed no preventative association of antidepressant treatment in pregnancy on reporting frequent SHI postpartum, relative to never/hardly ever SHI. Moreover, in a population of women with antenatal depression/anxiety, there also was no preventative association between past antidepressant treatment in pregnancy and reporting frequent SHI in the postpartum year.

The results of manuscript provide an advance in current knowledge. All conclusions are justified and supported by the results. The data and analyses are presented appropriately. Reviewers described their extensive search for studies. It is hard to find the weaknesses of this work, however I’ve some issues with antidepressant treatment characteristic.

  • In subsection 2. Self-reported depression and anxiety, and antidepressant treatment there is mention about ATC classification, however in the other parts of manuscript it does not exist.
  • Table 2 “SSRIs were the most commonly taken antidepressants, followed by SNRIs”. Please provide more information about SNRIs

Minor:

Table 1 – maternal age (years) - does not match the line

Author Response

Comment 1: The paper described the study to estimate the preventative association of antidepressants during pregnancy on postpartum self-harm ideation (SHI), using the Multinational Medication Use in Pregnancy Study. Frequency of SHI was measured via the Edinburgh Postnatal Depression Scale (EPDS). A sample of mothers who reported depression and/or anxiety during pregnancy (n=187) were included and mothers reported their antidepressant use in pregnancy retrospectively. The results of the study showed no preventative association of antidepressant treatment in pregnancy on reporting frequent SHI postpartum, relative to never/hardly ever SHI. Moreover, in a population of women with antenatal depression/anxiety, there also was no preventative association between past antidepressant treatment in pregnancy and reporting frequent SHI in the postpartum year.

The results of manuscript provide an advance in current knowledge. All conclusions are justified and supported by the results. The data and analyses are presented appropriately. Reviewers described their extensive search for studies. It is hard to find the weaknesses of this work, however I’ve some issues with antidepressant treatment characteristic.

In subsection 2. Self-reported depression and anxiety, and antidepressant treatment there is mention about ATC classification, however in the other parts of manuscript it does not exist.

Reply 1: Thank you for the overall positive feedback on our work. We have defined what ATC codes were considered in the definition of antidepressant exposure in the Methods; however, we feel that it will be overwhelming for the reader to repeat these ATC codes throughout the manuscript when they have been clearly defined in the methods. Most clinicians are familiar with the nomenclature SSRI / SNRI rather than ATC codes. We have added the following text as footnotes for all tables and figure 2: “Antidepressants included all selective serotonin re-uptake inhibitors (SSRIs, ATC code N06AB), serotonin and norepinephrine reuptake inhibitor (SNRI, ATC codes N06AX, venlafaxine and duloxetine), tricyclics (TCA, ATC code N06AA), and unspecified antidepressant (ATC code N06A-).”

Comment 2: Table 2 “SSRIs were the most commonly taken antidepressants, followed by SNRIs”. Please provide more information about SNRIs

Reply 2: We have amended the Results section by adding the following: “Venlafaxine was the predominant SNRI (n=9, 4.8%).”

Comment 3: Table 1 – maternal age (years) - does not match the line

Reply 3: Table 1 is correct in relation to maternal age data. We believe the table was split when the pdf file was built, and so the table headings were by chance repeated over the age variable. This issue will be fixed during the article proof. 
